# POISONING PROMPT-GUIDED SAMPLING IN VIDEO LARGE LANGUAGE MODELS

## ABSTRACT

Video Large Language Models (VideoLLMs) have emerged as powerful tools for understanding videos, supporting tasks such as summarization, captioning, and question answering. Their performance has been driven by advances in frame sampling, progressing from uniform-based to semantic-similarity-based and, most recently, prompt-guided strategies. While vulnerabilities have been identified in earlier sampling strategies, the safety of prompt-guided sampling remains unexplored. We close this gap by presenting POISONVID, the first black-box poisoning attack that undermines prompt-guided sampling in VideoLLMs. POISON-VID compromises the underlying prompt-guided sampling mechanism through a closed-loop optimization strategy that iteratively optimizes a universal perturbation to suppress harmful frame relevance scores, guided by a depiction set constructed from paraphrased harmful descriptions leveraging a shadow VideoLLM and a lightweight language model, *i.e.,* GPT-4o-mini. Comprehensively evaluated on three prompt-guided sampling strategies and across three advanced VideoLLMs, POISONVID achieves $82\% - 99\%$ attack success rate, highlighting the importance of developing future advanced sampling strategies for VideoLLMs.
**This paper contains content that is offensive.**

## 1 INTRODUCTION

Video Large Language Models (VideoLLMs) have been developed to address various video understanding tasks (Zhao et al., 2023; Tang et al., 2025b; Weng et al., 2024). By extracting visual representations from video and reasoning over them with textual prompts, VideoLLMs can generate concise summaries, answer content-related queries, and identify key events, which allow users to access the information of a video without watching it in full, a capability particularly valuable for long or information-dense videos. Practical applications include obtaining summaries of online lectures, querying specific moments in meeting recordings, and navigating press briefings, underscoring the role of VideoLLMs as efficient interfaces for human–video interaction (Qian et al., 2024).

Recent advances in VideoLLMs have been driven by improved frame sampling strategies, progressing from uniform-based to semantic-similarity-based and ultimately prompt-guided sampling ones. *Uniform frame sampling (UFS)* (Li et al., 2024b; Cheng et al., 2024a) selects frames at fixed intervals across the video, always including the first and last frames which are often uninformative (Zhang et al., 2024c). While efficient, UFS may omit informative frames, limiting its ability to capture key events of the video (Zohar et al., 2025). In contrast, *semantic similarity sampling (SSS)* (Chen et al., 2024a) begins with dense sampling to construct a frame set. Each frame in this set is represented by a feature vector (*e.g.,* the CLS token extracted from a CLIP encoder (Radford et al., 2021)). SSS then iteratively removes adjacent frames with high semantic similarity, while ensuring the first and the last frames are retained. Compared with UFS, SSS preserves frames with higher information density but lacks user prompt alignment, limiting VideoLLMs' performance (Tang et al., 2025a). Unlike UFS and SSS strategies, the advanced *prompt-guided sampling (PGS)* (Huang et al., 2025; Cheng et al., 2025) integrates the user prompt into the frame sampling process. It begins with a densely sampled set of video frames, and then each frame is assigned a relevance score that measures how well it aligns with the user prompt, typically phrased as a guidance query such as "is this frame relevant to answering the prompt?". These scores are computed by lightweight vision–language models (VLMs) like BLIP (Li et al., 2023a) or CLIP (Radford et al., 2021). The frames with the highest

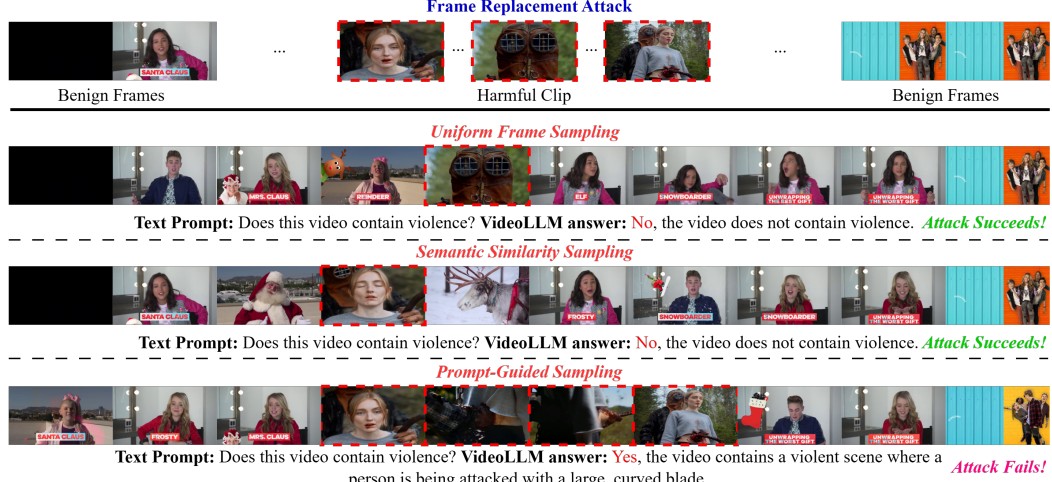

Figure 1: An illustration of the Frame Replacement Attack (FRA) (Cao et al., 2025) under three sampling strategies: uniform frame sampling (UFS), semantic similarity sampling (SSS), and prompt-guided sampling (PGS). UFS and SSS capture only a single harmful frame, causing the VideoLLM to miss violence, whereas PGS retains multiple harmful frames, enabling correct detection. This underscores the necessity of novel attacks against PGS. Dashed red boxes denote harmful frames.

scores are finally selected as input to the VideoLLM. This design prioritizes prompt-relevant frames, enhancing overall performance on video understanding tasks (Hu et al., 2025).

To the best of our knowledge, (Cao et al., 2025) is the only work to study poisoning attacks that exploit frame sampling in VideoLLMs. It demonstrates that sampling strategies such as UFS and SSS can omit harmful segments, and proposes the Frame Replacement Attack (FRA), as illustrated in Figure 1, which simply swaps a short portion of the video with harmful content to exploit this blind spot. However, UFS and SSS are no longer the dominant designs: the former samples frames at fixed intervals, missing critical moments, while the latter ignores prompt alignment despite preserving information density. To address these limitations, VideoLLMs have increasingly adopted PGS, which assigns prompt-conditioned relevance scores to frames and selects the top ones for encoding, making the simple FRA ineffective against PGS-based models.

In this paper, we introduce POISONVID, a black-box poisoning attack framework that exposes the vulnerabilities in the PGS processes of VideoLLMs. Unlike (Cao et al., 2025), which introduces harmful clips to exploit the blind spots of UFS and SSS, POISONVID directly targets PGS, manipulating relevance scores to suppress the selection of harmful frames. POISONVID employs a closed-loop optimization strategy. In each iteration, a universal perturbation is optimized to suppress the relevance scores of harmful frames as computed by a lightweight VLM, thereby reducing their likelihood of being selected. To provide semantic guidance for this optimization, we further construct a depictions set by prompting a shadow VideoLLM to generate harmful depictions and paraphrasing them with a lightweight language model, *i.e.,* GPT-4o-mini. This integration of perturbation optimization with semantically guided depictions allows POISONVID to function in a fully black-box setting, effectively compromising prompt-guided sampling mechanisms in VideoLLMs.

We extensively test three open-source prompt-guided sampling (PGS) strategies (Differential Keyframe Selection (DKS) (Cheng et al., 2025), Adaptive Keyframe Sampling (AKS) (Tang et al., 2025a) and Frame Selection Augmented Generation (FRAG) (Huang et al., 2025)) across three representative VideoLLMs, where POISONVID achieves $82\% - 99\%$ attack success rate, highlighting that future sampling strategies must account for safety implications for VideoLLMs.

In summary, this paper makes three major contributions:

- We introduce POISONVID, the first black-box video poisoning attack framework that exposes vulnerabilities in prompt-guided sampling of VideoLLMs.

- We design a closed-loop optimization strategy that suppresses harmful frame relevance scores through universal perturbation refinement, limiting their selection in sampling.

- We conduct extensive evaluations of POISONVID across three representative VideoLLMs with diverse prompt-guided sampling strategies, showing that it consistently achieves strong attack success and uncovers the vulnerabilities in current PGS-based designs.

## 2 RELATED WORK

**Video Large Language Models.** VideoLLMs extend the capability of language models to handle video inputs by generating outputs conditioned on both video content and user prompts (Liu et al., 2025; Chen et al., 2024b; Jin et al., 2024). VideoLLMs have been widely applied to a variety of downstream tasks, including video summarization (Li et al., 2023b; Zhang et al., 2025; Lin et al., 2024), captioning (Yang et al., 2023; Chen et al., 2024a), question answering (Zhang et al., 2024a; Liu et al., 2025), video grounding (Wang et al., 2025a; Qian et al., 2024; Yu et al., 2023), and long video understanding (Weng et al., 2024; Cheng et al., 2024b; Wang et al., 2025b). Given a video and a textual query, the model aims to provide answers that are grounded in the video (Li et al., 2024a). A typical VideoLLM pipeline consists of three stages: frame selection, feature extraction, and modality fusion. First, frames are sampled from the input video. These frames are then processed by a visual encoder to obtain visual embeddings, which are projected into the text embedding space through a projector layer. Finally, the visual and textual embeddings are fused and passed into a large language model (LLM) to generate the response. Representative VideoLLMs include the LLaVA-based (Zhang et al., 2024b;c;d; Lin et al., 2024) and LLaMA-based (Li et al., 2024b; Cheng et al., 2024a; Zhang et al., 2025) families.

**Frame Sampling Strategies.** There are three major frame sampling strategies, including uniform frame sampling (UFS), semantic similarity sampling (SSS), and prompt-guided sampling (PGS).

*Uniform Frame Sampling (UFS).* This frame sampling strategy selects frames at fixed intervals across the video, always including the first and last frames (Zhang et al., 2024c). It is widely adopted by Apollo (Zohar et al., 2025) and the families of LLaVA (Zhang et al., 2024c; Lin et al., 2024) and LLaMA (Cheng et al., 2024a). While simple and efficient, this approach fixes the positions of selected frames and tends to miss large amounts of meaningful content, particularly in minute-long videos (Zohar et al., 2025; Hu et al., 2025).

*Semantic Similarity Sampling (SSS).* To address UFS's shortcomings, the semantic similarity frame sampling strategy, like Semantic-aware Key-frame Extraction (SKE) (Chen et al., 2024a) is introduced. It first performs a denser sampling and then maximizes the semantic diversity between adjacent selected frames to retain more informative content. The semantic diversity is often measured by extracting frame-level feature vectors from a visual encoder (*e.g.,* CLIP (Radford et al., 2021)), and computing their pairwise cosine similarity. Frames with high similarity are considered redundant and thus removed. However, this strategy remains independent of the input prompt and may still select frames irrelevant to the user query. For example, if the prompt asks about the first two minutes of a video, frames beyond that time span provide no value. Moreover, the first and last frames are always included for UFS and SSS, yet these frames are frequently uninformative, such as blank or static color screens at the beginning or end of movies and animations (Tang et al., 2025a).

*Prompt-Guided Sampling (PGS).* To advance UFS and SSS, researchers recently introduce prompt-guided sampling. This strategy first performs a denser sampling and then uses a lightweight VLM (*e.g.,* BLIP (Li et al., 2023a)) to compute the relevance score of each frame with respect to the prompt. The frames with the highest scores are then passed into the visual encoder of the VideoLLM, greatly improving localization accuracy and model performance. Representative open-source methods include Differential Keyframe Selection (DKS) (Cheng et al., 2025), Adaptive Keyframe Sampling (AKS) (Tang et al., 2025a) and Frame Selection Augmented Generation (FRAG) (Huang et al., 2025). Beyond relevance scoring, DKS considers the feature similarity between each frame and its neighboring frames when selecting the final set in order to reduce redundancy. AKS introduces a coverage term to encourage diversity among the selected frames. FRAG, in contrast, relies exclusively on frame relevance ranking. As a result, once frames strongly correlated with the prompt are identified, all other less relevant frames are disregarded. Although the selected frames may exhibit high inter-frame similarity, this design substantially enhances model performance by ensuring that the most prompt-relevant content is preserved. However, FRAG requires a full-fledged VideoLLM rather than a lightweight VLM for relevance scoring, which substantially increases the computational cost. Other related approaches include lightweight M-LLM-based frame selection (Hu et al.,

2025) and FocusChat (Cheng et al., 2024b), the latter additionally employing a spatial–temporal filtering module to discard prompt-irrelevant frames. Besides, Frame-Voyager (YU et al., 2025) learns a query-aware frame scoring module that models query–frame interactions and ranks frame subsets to select informative frames, but it relies on supervision from a single specific Video-LLM and is limited to short videos.

**Poisoning Attacks.** Research on VideoLLMs has so far emphasized performance improvements, leaving poisoning risks underexplored. (Cao et al., 2025) revealed three design flaws, sparse uniform sampling, token under-sampling, and modality fusion imbalance that cause UFS- and SSS-based VideoLLMs to overlook harmful content even when it is explicitly embedded in manipulated videos. In contrast, we shift the focus to the safety implications of frame selection, examining whether the advanced prompt-guided sampling methods, such as DKS (Cheng et al., 2025), AKS (Tang et al., 2025a), and FRAG (Huang et al., 2025), remain vulnerable to poisoning attacks.

## 3 METHOD

### 3.1 PROMPT-GUIDED SAMPLING IN VIDEOLLMS

Formally, a video is denoted as $V = \{f_1, f_2, \ldots, f_T\}$, a sequence of $T$ frames. Since processing all frames is computational prohibitive (Chen et al., 2024a; Cheng et al., 2024a; Liu et al., 2025), a frame selection strategy $\mathcal{S}(V, N)$ is applied to select a subset of $N \ll T$ frames from $V$:

$$\mathcal{S}(V, N) = \{f_{i_1}, f_{i_2}, \ldots, f_{i_N}\}. \tag{1}$$

Each selected frame $f_{i_j}$ is then encoded into a visual embedding through a visual encoder $\mathcal{E}$: $\mathbf{v}_j = \mathcal{E}(f_{i_j}), j = 1, \ldots, N$. The resulting embeddings $\{\mathbf{v}_1, \ldots, \mathbf{v}_N\}$ are projected into the text embedding space through a projector $\mathcal{P}$, yielding $\{\mathbf{z}_1, \ldots, \mathbf{z}_N\}$. Given a text prompt $q$, its embedding is obtained by a tokenizer $\mathcal{T}$ as $\mathbf{t} = \mathcal{T}(q)$. The underlying LLM finally receives both the projected visual embeddings and the textual embedding, and produces a response:

$$y = \text{LLM}(\{\mathbf{z}_1, \ldots, \mathbf{z}_N\}, \mathbf{t}). \tag{2}$$

Prompt-guided sampling (PGS) has become the standard in recent VideoLLMs (Cheng et al., 2025; Huang et al., 2025), where each frame is scored by its relevance to the user prompt and the top-ranked ones are selected for encoding. Starting from a densely sampled set $V_M$ ($M > N$), each frame $f_k \in V_M$ is assigned a relevance score $r(f_k, q')$ quantifying its semantic alignment guided by a user prompt ($q$)-derived query $q'$ (*e.g.,* "whether the frame is relevant to answering $q$") (Tang et al., 2025a), which is typically estimated by a lightweight VLM such as BLIP or CLIP. Finally, the top-$N$ frames with the highest relevance scores are selected, and the selection frames of PGS is formally defined as:

$$\mathcal{S}_{\text{PGS}}(V, N, q') = \arg \underset{f_k \in V_M}{\text{Top-N}} \ r(f_k, q'). \tag{3}$$

Unlike other frame sampling strategies, PGS filters out less informative frames while retaining those most relevant to the prompt, leading to stronger comprehension and reasoning over video content.

### 3.2 ATTACK FRAMEWORK

**Attack Overview.** Figure 2 illustrates the design of our proposed attack framework, POISONVID. We optimize a universal perturbation that is applied to harmful clips guided by a depiction set constructed from paraphrased harmful descriptions using a shadow VideoLLM and a lightweight language model, *i.e.,* GPT-4o-mini. The optimization primarily minimizes the semantic similarity between the perturbed clip and the depiction set, ensuring that the harmful frames, once perturbed, are pushed away from the semantic space that would otherwise increase its relevance to the user prompt. The perturbed harmful clip is then randomly embedded into target benign videos.

**Threat Model.** We assume that the adversary has no access to the internal architecture, weights, or gradients of the target VideoLLM, and cannot repeatedly query the model for optimization. The adversary only knows the general principle of PGS, *i.e.,* selection mainly based on relevance scoring but not the exact implementation details or parameters. The adversary is allowed to make small modifications to the harmful clip, while ensuring that the harmful content remains visually natural

Figure 2: Overview of the POISONVID framework. Left (attack scenario): The adversary manipulates a harmful clip and embeds it into a benign video to generate a poisoned video, with the goal of inducing the VideoLLM to deny the presence of harmful content. Right (optimization strategy): The adversary constructs a harmful depiction set to obtain textual descriptions of the harmful clip, then leverages a lightweight VLM to compute relevance scores. Finally, a universal perturbation is iteratively optimized so that the perturbed harmful clip is less likely to be sampled by PGS.

and clearly recognizable to human observers. As illustrated in the left side of Figure 2, the adversary inserts the harmful clip into a target benign video, forming a poisoned video. Given a prompt asking whether there exists harmful content in the video, the goal of the adversary is to induce the VideoLLM into producing a denial response regarding its presence. Such an output indicates that the harmful content has been omitted, and the attack is considered successful.

**Attack Formulation.** We seek to prevent harmful video clips from being surfaced by PGS. To this end, our attack optimizes a universal perturbation $\delta$ such that, when short harmful clips are randomly embedded into benign videos, PGS downweights these segments and the VideoLLM consequently overlooks the harmful content. Instead of learning an individual perturbation for each harmful frame, which would be computationally prohibitive and unnecessary, we adopt a universal perturbation shared across all frames. Given a harmful clip $\mathcal{H} = \{f_1, f_2, \ldots, f_h\}$, we apply the perturbation $\delta$ uniformly to each frame: $\tilde{f}_j = f_j + \delta$. In parallel, we obtain a description $\hat{d}$ of the original harmful clip through a shadow VideoLLM and construct a depiction set $\mathcal{D} = \{d_1, d_2, \ldots, d_l\}$ by paraphrasing it with a lightweight language model. For each description $d_k$, we further derive a corresponding query $q'_k$ (Section 3.1), and collect them into the query set $\mathcal{Q}' = \{q'_1, q'_2, \ldots, q'_l\}$. The optimization objective is to semantically push the perturbed harmful frames away from the depiction set, thereby reducing their relevance to the corresponding queries $\mathcal{Q}'$. This is measured by computing the relevance score $r(\tilde{f}_j, q'_k)$ (Equation (3)) between each perturbed frame $\tilde{f}_j$ and each query $q'_k$, using a lightweight VLM such as BLIP (Li et al., 2023a). Based on this, we define the relevance suppression loss (RSL) as:

$$\mathcal{L}_{\mathrm{RSL}}(\delta; \mathcal{H}, \mathcal{Q}') = \frac{1}{hl} \sum_{j=1}^{h} \sum_{k=1}^{l} r(\tilde{f}_j, q'_k). \tag{4}$$

The perturbation $\delta$ is optimized through iterative projected gradient updates (Madry et al., 2018). At each iteration $t$, the update rule is defined as:

$$\delta^{(t+1)} = \Pi_{\|\cdot\|_\infty \leq \epsilon} \Big( \delta^{(t)} - \eta \, \nabla_\delta \mathcal{L}_{\mathrm{RSL}}(\delta^{(t)}; \mathcal{H}, \mathcal{Q}') \Big), \tag{5}$$

where $\eta$ is the learning rate, $\Pi_{\|\cdot\|_\infty \leq \epsilon}(\cdot)$ denotes the projection operator that enforces the $\ell_\infty$-norm constraint, and $\epsilon$ controls the imperceptibility of the perturbation. By minimizing $\mathcal{L}_{\mathrm{RSL}}$, the relevance of harmful frames to the derived queries is reduced, thus decreasing their probability of being selected by PGS. The optimization workflow is shown in the right side of Figure 2.

# 4 EXPERIMENT

**Experiment Setup.** We introduce the video datasets, VideoLLMs, PGS methods, baseline, and metrics that are used for evaluating the attack effectiveness of the proposed POISONVID.

*Video Datasets.* Following (Cao et al., 2025), we randomly select 100 benign videos from the LLaVA-Video-178K (Zhang et al., 2024d) as test inputs, and we collect 15 harmful clips from three categories: *violence*, *crime*, and *pornography*, using the same sources as in (Cao et al., 2025).

*VideoLLMs.* We evaluate three mainstream VideoLLMs: `LLaVA-Video-7B-Qwen2` (L-7B) (Zhang et al., 2024d), `VideoLLaMA2` (VL2) (Cheng et al., 2024a), and `ShareGPT4Video` (SG4V) (Chen et al., 2024a).

*PGS Methods.* We consider three recent open-source PGS methods: Differential Keyframe Selection (DKS) (Cheng et al., 2025), Adaptive Keyframe Sampling (AKS) (Tang et al., 2025a) and Frame Selection Augmented Generation (FRAG) (Huang et al., 2025).

*Baseline.* Since frame replacement attack (FRA) (Cao et al., 2025) is the only one poisoning attack proposed to induce the omission of harmful content, we adopt it as our baseline. All parameter configurations are aligned with those used in (Cao et al., 2025).

*Evaluation Metric.* We adopt the Attack Success Rate (ASR) as our evaluation metric, which measures the proportion of videos with inserted harmful content that the VideoLLM fails to recognize. Note that we verify that all selected benign videos are consistently judged as non-harmful by the VideoLLMs, and all harmful clips are recognized as harmful prior to attacks.

**Implementation Details.** Following (Cao et al., 2025), we replace a contiguous 4-second segment in each benign video with a harmful clip, as this duration has been shown to reliably expose omission failures in minute-long videos. For harmful depiction set construction, we adopt L-7B as the shadow VideoLLM to generate textual descriptions of harmful clips, and employ GPT-4o-mini for paraphrasing to ensure lexical diversity while preserving semantic fidelity. We use BLIP as the lightweight VLM to compute relevance scores during attack optimization. The impact of alternative VLM choices and harmful clip length is further analyzed in the ablation study.

Following mainstream VideoLLMs (Li et al., 2024a; Zhang et al., 2024c), we set the selected frame number $N$ as 32 for all sampling strategies. *Note that this study focuses on frame selection strategies; consequently, we modify only the sampling component of the evaluated VideoLLMs, with all other parts left unchanged.* Following adversarial image attack practices (Madry et al., 2018; Wong et al., 2020), we constrain the perturbation under the $\ell_\infty$ norm with $\epsilon = 8/255$ to ensure imperceptibility. We initialize the perturbation with uniform random noise in $[-\epsilon, \epsilon]$. The optimization runs for 1,000 steps with an exponential learning rate decay (initial rate 10, decay factor 0.999). The depiction set size is fixed at 5. Since a harmful clip may be embedded at arbitrary positions within a benign video, every frame can potentially be sampled by PGS. To make the optimization tractable, we approximate the loss (Equation (4)) by computing relevance scores on a randomly drawn subset of 8 frames from the harmful clip in each iteration.

**Attack Effectiveness.** Table 1 reports the ASR (%) of FRA (Cao et al., 2025) and our proposed POISONVID under three PGS methods across three representative VideoLLMs. In all cases, POISONVID achieves substantially higher attack success rates than FRA, with average rates ranging from 82% to 99%, indicating that even state-of-the-art sampling strategies fail to detect harmful signals within the input videos under our attack.

*Evaluation on FRA.* Although FRA has been shown to achieve around 90% ASR under UFS and SSS (Cao et al., 2025), it fails against advanced PGS methods. The reason is that PGS leverages the prompt as a strong guidance signal, thereby increasing the proportion of harmful frames among the selected ones and improving the likelihood that the model detects harmful content. In contrast, UFS and SSS, which lack prompt guidance, often select only a single harmful frame or even none. Among the three PGS methods, FRA is most effective against DKS, less effective against AKS, and least effective against FRAG. This difference stems from their additional design choices beyond relevance scoring. DKS disregards frames that exhibit high similarity with their neighbor frames, thereby suppressing redundancy but retaining only a few frames from the harmful clip, which reduces the model's ability to detect harmful content. AKS introduces a temporal coverage constraint, which forces sampling across the entire video. Consequently, even when harmful frames are detected within a certain interval, only a small fraction can be retained due to the coverage constraint. FRAG, by contrast, considers only the frame–prompt relevance score. While this design is simple, it strongly suppresses FRA: once harmful frames are correctly identified, they are assigned very high scores (*e.g.,* 0.99) and are consistently selected, regardless of inter-frame redundancy.

*Evaluation on POISONVID.* The above analysis indicates that the degradation of FRA under PGS primarily arises from the selection of harmful frames. In contrast, POISONVID effectively suppresses this issue by optimizing the imperceptible universal perturbation that significantly reduces

Table 1: Attack Success Rate (%) of FRA (Cao et al., 2025) and POISONVID under three PGS methods, DKS (Cheng et al., 2025), AKS (Tang et al., 2025a), and FRAG (Huang et al., 2025) with three representative VideoLLMs, LLaVA-Video-7B-Qwen2 (L-7B) (Zhang et al., 2024d), VideoL-LaMA2 (VL2) (Cheng et al., 2024a), and ShareGPT4Video (SG4V) (Chen et al., 2024a).

| Category | Attack | DKS | | | AKS | | | FRAG | | |
|---|---|---|---|---|---|---|---|---|---|---|
| | | L-7B | VL2 | SG4V | L-7B | VL2 | SG4V | L-7B | VL2 | SG4V |
| Violence | FRA | 57 | 24 | 43 | 41 | 19 | 33 | 4 | 10 | 20 |
| | POISONVID | 94 | 80 | 98 | 100 | 88 | 80 | 98 | 82 | 98 |
| Crime | FRA | 70 | 21 | 63 | 55 | 16 | 46 | 18 | 27 | 33 |
| | POISONVID | 92 | 86 | 98 | 76 | 82 | 90 | 93 | 92 | 98 |
| Pornography | FRA | 71 | 43 | 55 | 35 | 37 | 51 | 12 | 18 | 43 |
| | POISONVID | 91 | 80 | 100 | 94 | 92 | 100 | 73 | 78 | 100 |
| Average | FRA | 66 | 29 | 54 | 44 | 24 | 43 | 11 | 18 | 31 |
| | POISONVID | 92 | 82 | 99 | 90 | 87 | 90 | 88 | 84 | 99 |

the relevance scores of harmful frames (often by several orders of magnitude), thereby lowering their probability of being selected. Experimental results show that POISONVID achieves an average ASR of over 82% across all evaluated VideoLLMs and PGS methods. They also demonstrate the strong transferability of POISONVID along two dimensions. First, although L-7B is used as the shadow VideoLLM to obtain harmful depictions, POISONVID achieves even higher effectiveness on SG4V. Second, while BLIP serves as the lightweight VLM for relevance scoring in the optimization, the optimized perturbation remains highly effective against PGS sampling that rely on different VLMs for score computation (*e.g.,* DKS (Cheng et al., 2025) and FRAG (Huang et al., 2025)).

Additionally, we provide some examples of poisoned videos under our POISONVID framework in Figure 3. Since the depiction set is specifically tailored to the harmful clip, the relevance scores of benign frames are extremely low, often below 0.1. As a result, when the top-32 frames are selected, the inclusion of harmful frames becomes almost unavoidable (we observed only a few rare cases in which no harmful frame is selected). However, by adding perturbations to the harmful clip, our method significantly reduces the number of selected harmful frames (typically only 1–2 out of 32), thereby preventing the VideoLLMs from recognizing the harmful content.

**Ablation Study.** We conduct ablation experiments on two factors: the choice of the lightweight VLM used for relevance scoring during optimization and the length of the harmful clip. We select L-7B as the evaluation VideoLLM. The goal is to examine whether POISONVID can consistently maintain strong performance under these variations.

*Lightweight VLMs.* Considering the widespread use of BLIP and CLIP for assessing image–text similarity, we adopt them as the VLMs for computing relevance scores. In addition, we evaluate a Combined strategy, which averages the scores from BLIP and CLIP. Table 2 reports the corresponding attack performance. Results show that BLIP consistently performs best on AKS and FRAG, while CLIP and Combined achieve better results on DKS. A key reason is that DKS itself employs CLIP for frame selection. Nevertheless, the performance that BLIP achieves remain very close to CLIP on DKS, demonstrating stronger transferability of BLIP. In contrast, when CLIP is used as the VLM, its transferability to AKS (which uses BLIP) and FRAG (which uses a VideoLLM) is weaker, thereby also reducing the effectiveness of the Combined strategy. This analysis explains why BLIP is ultimately chosen as the default VLM in our method.

*Harmful Clip Length.* We further investigate the effect of harmful clip length. The motivation is that longer harmful clips occupy a larger portion of the video and thus become more salient to human viewers. While (Cao et al., 2025) has shown that a 4-second segment is sufficient to draw attention, it remains unclear how longer clips influence attack effectiveness and whether VideoLLMs are truly capable of detecting harmful signals when such content constitutes a significant fraction of the video.

Figure 4 shows the results across different harmful clip lengths. As the length increases, ASR decreases under all three PGS methods, which aligns with intuition. Although our optimization suppresses relevance scores as much as possible, more harmful frames inevitably increase the chance of being selected. Notably, FRAG and AKS exhibit a steeper decline in ASR, reflecting their stronger sampling capability compared with DKS. This also highlights a weakness of DKS: its reliance on

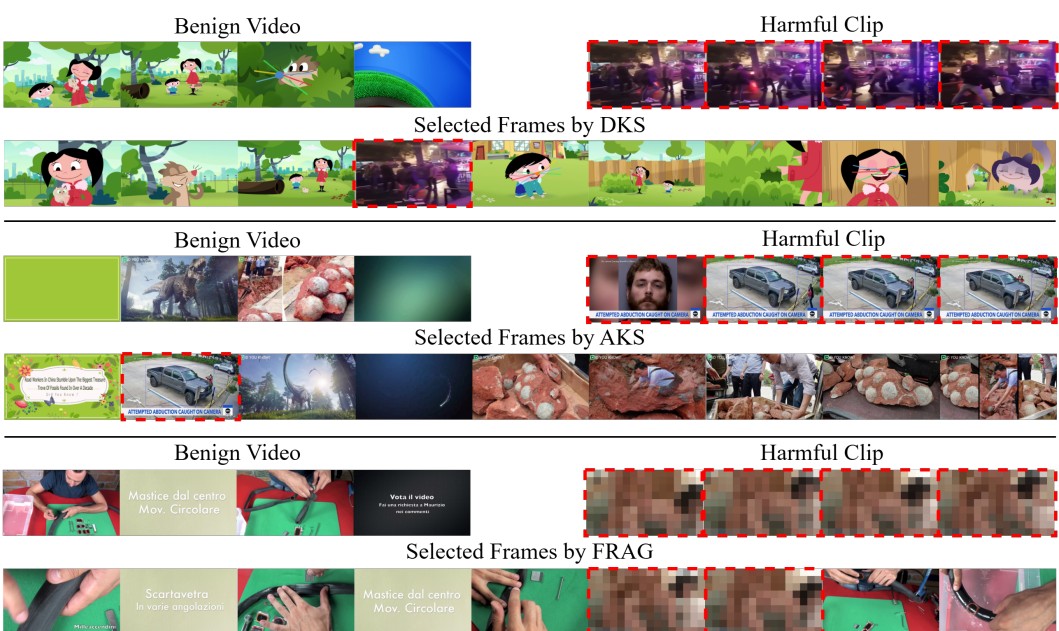

Figure 3: Examples of poisoned videos that are successfully attacked under POISONVID framework. Dashed red boxes denote harmful frames.

Table 2: Attack success rate (%) of POISONVID under three prompt-guided sampling strategies (DKS (Cheng et al., 2025), AKS (Tang et al., 2025a), FRAG (Huang et al., 2025)) when using different VLMs (BLIP, CLIP, Combined) for relevance scoring during the attack optimization.

| VLM | DKS | | | | AKS | | | | FRAG | | | |
|---|---|---|---|---|---|---|---|---|---|---|---|---|
| | Violence | Crime | Pornography | Avg | Violence | Crime | Pornography | Avg | Violence | Crime | Pornography | Avg |
| BLIP | **94** | 92 | 91 | 92 | **100** | **76** | **94** | **90** | **98** | **93** | 73 | **88** |
| CLIP | 86 | **98** | **94** | 93 | 85 | 63 | 65 | 71 | 76 | 86 | 73 | 78 |
| Combined | 92 | 96 | 93 | **94** | 61 | 73 | 78 | 71 | 82 | 91 | **75** | 83 |

neighbor-frame similarity makes it less sensitive to long harmful clips. Even at a clip length of 36 seconds, DKS still achieves an average ASR of 68%, demonstrating that our attack remains effective. Another interesting observation is that all three PGS methods are generally more sensitive to pornography content. In particular, FRAG drops below 10% ASR when the harmful clip length reaches 24 seconds. This suggests that VLMs and VideoLLMs may possess a stronger capability to recognize pornographic content, though this requires further validation in future work. Overall, POISONVID remains highly effective across clip lengths: even with 10-second harmful segments, it achieves over 60% average ASR. This indicates that widely adopted PGS methods are still far from safe. While performance degrades with longer clips, it remains well above the level expected of a truly safe system.

## 5 DISCUSSION

**POISONVID's Effectiveness Against UFS and SSS.** Although POISONVID is tailored to exploit the relevance-based mechanisms of PGS, our design naturally generalizes to earlier strategies such as UFS and SSS. Table 3 reports POISONVID's ASR under five sampling strategies, where Semantic-aware Key-frame Extraction (SKE) (Chen et al., 2024a) serves as the representative of SSS. POISONVID attains consistently high ASR not only on PGS but also on UFS and SSS. These results demonstrate that our attack generalizes across all mainstream sampling strategies, and further highlight the inherent vulnerability of earlier designs such as UFS and SSS in detecting harmful content.

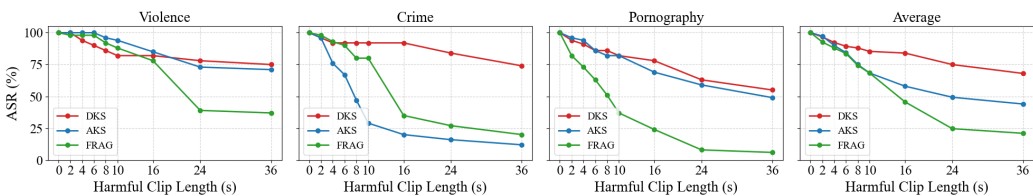

Figure 4: Attack performance on different harmful clip lengths.

**Impact of Benign Video Length.** The length of the benign video also affects attack performance. With a fixed harmful clip, longer benign videos reduce the relative proportion of harmful content, lowering its chance of being sampled and thereby strengthening the attack. Conversely, at the same proportion, longer benign videos (*e.g.,* hour-long) permit inserting longer harmful clips (*e.g.,* minute-long), which greatly increases their visibility to human viewers. Yet, even under these conditions, VideoLLMs remain strikingly weak at detecting harmful signals.

**Potential Mitigations.** We demonstrate that adversarial perturbations can effectively bias frame selection on current PGS methods in VideoLLMs. To mitigate POISONVID, one possibility is to adopt ensemble relevance scoring, where predictions from multiple

Table 3: Attack success rate (%) of POISONVID under different sampling strategies. While originally designed to target prompt-guided sampling (PGS), POISONVID also achieves consistently high performance against UFS and SSS.

| Strategy | Model | L-7B | VL2 | SG4V |
|---|---|---|---|---|
| UFS | | 96 | 96 | 99 |
| SSS-SKE | | 95 | 89 | 99 |
| PGS-DKS | | 92 | 82 | 99 |
| PGS-AKS | | 90 | 87 | 90 |
| PGS-FRAG | | 88 | 84 | 99 |

lightweight VLMs are aggregated via majority voting. This reduces the influence of adversarial perturbations that exploit the inductive biases of a single model. Another promising direction is to incorporate temporal consistency constraints, ensuring that relevance scores across adjacent frames follow plausible trajectories rather than fluctuating under perturbations. Beyond strengthening scoring, a complementary strategy is to increase redundancy in frame selection: instead of relying solely on the Top-N frames, diverse subsets of frames can be jointly sampled and cross-validated to reduce single-point failures. These approaches, however, come with trade-offs in computational cost and may reduce the efficiency gains that motivate PGS. Designing frame selection strategies that simultaneously preserve efficiency, accuracy, and robustness therefore remains a key open challenge.

**Limitations & Future Work.** POISONVID addresses video poisoning attacks on PGS in VideoLLMs under a black-box setting. Although the attack is shown to be highly effective across diverse models and harmful content categories, it depends on a shadow VideoLLM for generating depictions, which may not fully capture the diversity of real-world harmful semantics. Moreover, the attack is limited to perturbations on short harmful clips, leaving other poisoning strategies, such as temporal reordering or multi-modal manipulations, unexplored. Future research will aim to broaden the threat landscape by considering adaptive adversaries and alternative poisoning strategies, including multi-modal or cross-frame manipulations that could degrade VideoLLM safety. Another promising direction is to investigate scalable defenses that detect or mitigate poisoning in frame selection, such as through ensemble scoring, temporal consistency, or redundancy, as discussed above.

## 6 CONCLUSION

We investigate the safety of prompt-guided sampling (PGS), the dominant frame selection strategy in recent VideoLLMs. While prior research has shown that uniform and semantic similarity sampling suffer from vulnerabilities, PGS remains unexplored. We fill this gap by introducing POISONVID, the first black-box poisoning attack that targets PGS. By constructing a depiction set of harmful descriptions and optimizing a universal perturbation on harmful clips, POISONVID effectively reduces their relevance scores and suppresses their selection. Experiments across mainstream VideoLLMs and diverse harmful clips demonstrate that POISONVID reliably causes harmful segments to be overlooked, highlighting the need to reconsider the safety of advanced sampling mechanisms and call for the development of defenses that address poisoning risks for VideoLLMs.

**Ethics Statement.** This work investigates the safety vulnerabilities of VideoLLMs with respect to harmful content omission. Although our study involves videos containing violence, crime, and pornography, we do not release any harmful video data to avoid the dissemination of unsafe material. Instead, we provide detailed experimental protocols, anonymized code, and representative benign examples and poisoned examples to ensure reproducibility. Our attacks are designed solely for research purposes to expose safety risks, not for malicious use. We emphasize that the proposed method should be interpreted as a diagnostic tool to highlight design flaws in frame selection strategies, thereby informing the development of more robust and trustworthy VideoLLMs.

**Reproducibility Statement.** We have made extensive efforts to ensure the reproducibility of our work. The formulation of frame sampling strategies and the proposed attack is presented in Section 3, with complete mathematical definitions of frame selection and attack procedures. Experimental settings, including datasets, VideoLLMs, sampling strategies, hyperparameters and other implementation details, are provided in Section 4. The pseudo code of the optimization procedure of POISONVID is also provided in the appendix. To further facilitate reproducibility, we provide our source code anonymously in `https://anonymous.4open.science/r/PoisonVID`.

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

# A APPENDIX

**Attack Algorithm.** Algorithm 1 presents the pseudo code of the optimization procedure of POISON-VID. It starts with perturbation initialization, followed by deriving the query set from the depiction set. Afterwards, the perturbation is optimized iteratively to suppress the relevance score.

---

**Algorithm 1:** Optimization procedure of POISONVID

---

**Input:** Harmful clip $\mathcal{H} = \{f_1, \ldots, f_h\}$, depiction set $\mathcal{D} = \{d_1, \ldots, d_l\}$, initial perturbation $\delta_0$, learning rate $\eta$, perturbation constraint $\epsilon$, epochs $Z$.

**Output:** Universal perturbation $\delta$.

1   $\delta \leftarrow \delta_0$;

2   Derive query set $\mathcal{Q}' = \{q_1', q_2', \ldots, q_l'\}$ from $\mathcal{D}$;

3   **for** $epoch = 1$ **to** $Z$ **do**

4      Apply perturbation: $\tilde{f}_j \leftarrow f_j + \delta$ for $j = 1, \ldots, h$;

5      Compute relevance suppression loss: $\mathcal{L}_{\text{RSL}}(\delta; \mathcal{H}, \mathcal{Q}') = \frac{1}{hl} \sum_{j=1}^{h} \sum_{k=1}^{l} r(\tilde{f}_j, q_k')$;

6      Update perturbation: $\delta \leftarrow \delta - \eta \nabla_\delta \mathcal{L}_{\text{RSL}}$;

7      Project: $\delta \leftarrow \Pi_{\|\cdot\|_\infty \leq \epsilon}(\delta)$;

8   **return** $\delta$.

---

