# OpenReview forum: "Poisoning Prompt-Guided Sampling in Video Large Language Models"
_ICLR.cc/2026/Conference — ICLR 2026 Conference Withdrawn Submission_

### Official Review · Reviewer_gyYm · 2025-10-24

**Soundness:** 2
**Presentation:** 3
**Contribution:** 2
**Rating:** 2
**Confidence:** 4

**Summary:**

This paper introduces PoisonVID, an attack framework targeting PGS strategies in VideoLLMs. The attack works by optimizing a universal perturbation that reduces the relevance scores of harmful video frames, causing them to be excluded during the frame selection process. The authors construct a "depiction set" by using a shadow VideoLLM to generate descriptions of harmful content and paraphrasing them with GPT-4o-mini. These descriptions guide the optimization of perturbations that push harmful frames away semantically from what PGS would consider relevant. The method is evaluated on three VideoLLMs with three different PGS strategies, achieving 82-99% attack success rates. The work highlights vulnerabilities in current advanced sampling strategies for VideoLLMs.

**Strengths:**

- The paper effectively explains why existing attacks like FRA fail against PGS methods, establishing the need for a new approach. The progression from UFS to SSS to PGS is well explained.
- The authors test their method across multiple VideoLLMs (3 models), multiple PGS strategies (3 methods), and multiple harmful content categories (violence, crime, pornography), demonstrating breadth in their evaluation.

**Weaknesses:**

- The paper claims to be a black-box attack, but the method requires access to the VLM used in the PGS process during optimization. Since the VLM is part of the VideoLLM's frame sampling pipeline, accessing it means accessing a core component of the target system. This makes the threat model closer to gray-box than truly black-box.
- The paper does not clearly explain the practical scenario where this attack would be used. Who would deploy such attacks and why? What are the real-world implications? The motivation for studying this specific threat model needs better justification.
- The method appears to apply standard adversarial perturbation optimization from images to videos. The paper does not clearly explain why simpler attacks (like adding random Gaussian noise or adapting FGSM) would not work. They can be highly effective and achieve the same goal of the proposed method, but without complex optimization. Without comparisons to these simpler baselines, it's hard to assess whether the proposed optimization is necessary.

**Questions:**

- Can you provide more analysis on why pushing harmful frames away from the depiction set reduces their relevance scores? The relationship seems indirect: if the depiction set comprehensively spans the harmful semantic space, then pushing away from it might reduce harmfulness, which could explain the lower relevance scores. However, if the depiction set is small or incomplete, it's less clear why the relevance scores drop. An ablation study varying the depiction set size (e.g., 1, 5, 10, 20 descriptions) and analyzing how this affects both the embedding distances and the resulting relevance scores would help clarify this mechanism. Additionally, since this is a safety paper about VideoLLMs, exploring how the harmful content's position in the embedding space relates to the sampling behavior could provide valuable insights beyond the attack itself.
- The paper calls the perturbation "universal" but does it only applies to frames within a single harmful clip, not across different harmful clips. This usage of "universal" is different from standard adversarial machine learning terminology where universal perturbations work across different inputs. Does the same perturbation work for different harmful clips of the same/ different category (e.g., different violence clips)?
- How much time and computation does the optimization require? What is the practical overhead?
- The authors mention that PGS methods are "more sensitive to pornography content". However, this seems to depend on the prompt. If ask "Does this video contain pornography?" wouldn't the model detect it better? Can the authors clarify this observation?
- In lines 450-451, the authors state VideoLLMs are "strikingly weak at detecting harmful signals" in long videos. Is this weakness due to the VideoLLM's inherent difficulty with long videos, or is it because of PoisonVID attack's effectiveness? Can these two factors discussed separately?

---

### Official Review · Reviewer_Wu6B · 2025-10-30

**Soundness:** 2
**Presentation:** 3
**Contribution:** 2
**Rating:** 4
**Confidence:** 3

**Summary:**

The paper introduces POISONVID, the first black-box poisoning attack that targets prompt-guided frame sampling (PGS) in VideoLLMs. Instead of inserting overt harmful segments to exploit uniform or semantic sampling, POISONVID learns a universal, imperceptible perturbation that suppresses PGS relevance scores for harmful frames via a closed-loop optimization guided by a depiction set (paraphrased harmful descriptions produced with a shadow VideoLLM and a lightweight LLM). This causes harmful segments to be under-sampled and subsequently overlooked by downstream VideoLLMs. Across three open PGS methods (DKS, AKS, FRAG) and three VideoLLMs (LLaVA-Video-7B-Qwen2, VideoLLaMA2, ShareGPT4Video), POISONVID attains 82–99% attack success rate (ASR), substantially outperforming a Frame-Replacement baseline; ablations examine lightweight VLM choices (BLIP/CLIP) and harmful-clip length, and the discussion sketches potential mitigations (ensemble scoring, temporal consistency, redundancy).

**Strengths:**

- Empirical evidence: High ASR (82–99%) across 3× PGS and 3× VideoLLMs, with detailed comparisons to FRA and clear analyses of why PGS resists FRA but not POISONVID.

**Weaknesses:**

- Scope of harmful content & external dependencies: Experiments focus on three categories (violence, crime, pornography) and require a shadow VideoLLM plus GPT-4o-mini for depiction/paraphrase, raising questions about generality, cost, and potential bias.
- Attack surface confined to sampling: The method targets frame selection; it remains unclear how robust defenses at feature extraction, fusion, or generation stages would affect ASR in end-to-end pipelines. (Inferred from the pipeline focus.)
- Perceptual/operational realism underexplored: While perturbations are bounded and described as imperceptible, the paper lacks human perceptual validation and operational constraints (compression, re-encoding, platform filters) analyses.

**Questions:**

- What additional end-to-end defenses or pipeline changes (e.g., adversarially trained encoders, temporal fusion checks) would you expect to substantially reduce ASR, and should the paper evaluate POISONVID post-sampling robustness explicitly?


- What human-study or codec-robustness evaluations (e.g., H.264 re-encoding, bitrate downscaling, platform filters) are necessary to accept the claim that perturbations are truly imperceptible and durable in real-world video pipelines?


- Given reliance on a shadow VideoLLM and GPT-4o-mini, what audits or alternative construction protocols (open-source LLMs, diverse paraphrase sources) would convince you that results generalize beyond the current harmful categories and are not artifacts of teacher bias?

---

### Official Review · Reviewer_QXgc · 2025-10-31

**Soundness:** 2
**Presentation:** 3
**Contribution:** 2
**Rating:** 2
**Confidence:** 5

**Summary:**

This paper designs a black-box video poisoning attack framework that interferes with VideoLLMs’ ability to recognize harmful or policy-violating content. The goal is to diagnose and analyze the model’s vulnerability and diagnostic capability toward safety-sensitive content.

**Strengths:**

The proposed method is simple yet effective, featuring a closed-loop optimization strategy that suppresses the relevance scores of harmful frames through universal perturbation refinement, effectively reducing their chance of being selected during sampling.

**Weaknesses:**

1. The task setup appears relatively simple, mainly targeting safety-related cues such as violence detection. It is unclear whether the proposed attack generalizes to broader, open-ended VideoQA or reasoning tasks. Focusing only on specific “harmful” topics (e.g., violence, nudity) may limit the broader significance of the contribution.

2. I am curious about how the proposed attack performs on long-video VideoLLMs or closed-source models. Since the framework is presented as black-box, it would be valuable to see whether the attack remains effective in these more challenging and realistic settings.

3. The idea of using visual prompt–based perturbations to mislead models is not particularly novel. Similar mechanisms have been explored in previous adversarial or data poisoning works, so the technical originality of this paper remains questionable.

4. The paper could be improved. The motivation and background sections are overly lengthy, while the method is introduced relatively late. This makes the logical flow less clear and reduces readability.

**Questions:**

Please see the weaknesses.

---

### Official Review · Reviewer_htnF · 2025-11-01

**Soundness:** 2
**Presentation:** 3
**Contribution:** 2
**Rating:** 4
**Confidence:** 4

**Summary:**

This paper introduces POISONVID, the first black-box poisoning attack targeting prompt-guided frame sampling in Video Large Language Models (VideoLLMs). The method employs a closed-loop optimization to craft a universal perturbation that manipulates frame relevance scores, effectively misleading the sampling process. Using a depiction set derived from paraphrased harmful prompts and guided by a shadow VideoLLM plus a lightweight language model (GPT-4o-mini), POISONVID achieves 82%–99% attack success across multiple models and strategies. The results reveal a critical vulnerability in current prompt-guided sampling pipelines and underscore the need for robust and secure sampling mechanisms in future VideoLLMs.

**Strengths:**

1. The paper is well-written and easy to follow, with clear motivation and structure.

2. The topic is novel and timely — it’s the first work to propose a black-box poisoning attack (POISONVID) against prompt-guided sampling (PGS), directly addressing the key question of whether advanced sampling strategies are truly secure. The research focus is precise and forward-looking.

3. The experiments are comprehensive, covering three mainstream VideoLLMs (L-7B, VL2, SG4V), three representative PGS methods (DKS, AKS, FRAG), and three types of harmful content (violence, crime, pornography). The inclusion of traditional baselines (Uniform, SSS) makes the evaluation broad and convincing, effectively demonstrating the generality of the proposed attack.

**Weaknesses:**

1. The paper claims that UFS and SSS are no longer dominant designs; however, uniform sampling remains the default choice in many mainstream open-source VLMs such as Qwen-VL and InternVL, which limits the practical impact and applicability of this work.

2. Several other PGS-related approaches are not discussed, such as those integrating with agents [1,2] or motion cues [3].

3. The experimental evidence is not fully convincing. It’s unclear whether the authors trained models specifically for each PGS method, which is crucial for evaluating generalization. In addition, sampling only 100 videos seems too limited to ensure diversity in both length and content. Simply “following prior work” is not a sufficient justification here.

[1] VideoAgent: Long-Form Video Understanding with Large Language Model as Agent. ECCV24

[2] Videotree: Adaptive tree-based video representation for llm reasoning on long videos. CVPR25

[3] Flow4Agent: Long-form Video Understanding via Motion Prior from Optical Flow. ICCV25

**Questions:**

1. Does the perturbation generated by POISONVID exhibit cross-harm-type transferability? For example, can a perturbation optimized for “violent” clips also degrade the relevance of “sexual” clips? If not, which key components would need adjustment to improve such transferability?

2. In constructing the harmful description set, what specific paraphrasing strategy was used with GPT-4o-mini? For instance, was semantic similarity controlled (e.g., via BLEU-based filtering)? If a paraphrase deviates too much semantically from the original, could this mislead the direction of perturbation optimization?

---

### Note · Authors · 2025-11-12

I have read and agree with the venue's withdrawal policy on behalf of myself and my co-authors.